# Understanding how community antiretroviral delivery influences engagement in HIV care: a qualitative assessment of the Centralised Chronic Medication Dispensing and Distribution programme in South Africa

Jienchi Dorward ◆ ,[1,2] Lindani Msimango,[2] Andrew Gibbs ◆ ,[3,4] Hlengiwe Shozi,[2] Sarah Tonkin-Crine ◆ ,[1,5] Gail Hayward ◆ ,[1] Christopher C Butler ◆ ,[1] Hope Ngobese ◆ ,[6] Paul K Drain ◆ ,[7,8,9] Nigel Garrett ◆ [2,10]

For numbered affiliations see end of article.

**Correspondence to**
Dr Jienchi Dorward;
jienchi.dorward@phc.ox.ac.uk

## ABSTRACT

**Introduction** Providing antiretroviral therapy (ART) for millions of people living with HIV requires efficient, client-centred models of differentiated ART delivery. In South Africa, the Centralised Chronic Medication Dispensing and Distribution (CCMDD) programme allows over 1 million people to collect chronic medication, including ART, from community pick-up points. We aimed to explore how CCMDD influences engagement in HIV care.

**Methods** We performed in-depth interviews and focus group discussions with clients receiving ART and healthcare workers in Durban, South Africa. We analysed transcripts using deductive thematic analysis, with a framework informed by 'theories of practice', which highlights the *materialities*, *competencies*, *meanings* and *other life practices* that underpin clients' engagement in HIV care.

**Results** Between March 2018 to August 2018 we undertook 25 interviews and four focus groups with a total of 55 clients, and interviewed eight healthcare workers. The material challenges of standard clinic-based ART provision included long waiting times, poor confidentiality and restricted opening hours, which discouraged clients from engagement. In contrast, CCMDD allowed quicker and more convenient ART collection in the community. This required the development of new competencies around accessing care, and helped change the meanings associated with HIV, by normalising treatment collection. CCMDD was seen as a reward by clients for taking ART well, and helped reduce disruption to other life practices such as employment. At private pharmacies, some clients reported receiving inferior care compared with paying customers, and some worried about inadvertently revealing their HIV status. Clients and healthcare workers had to negotiate problems with CCMDD implementation, including some pharmacies reaching capacity or only allowing ART collection at restricted times.

**Conclusions** In South Africa, CCMDD overcame material barriers to attending clinics, changed the meanings associated with collecting ART and was less disruptive to other social practices in clients' lives. Expansion of community-based ART delivery programmes may help to facilitate engagement in HIV care.

**Trial registration number** STREAM study clinical trial registration: NCT03066128, registered February 2017.

## Strengths and limitations of this study

► This is one of the first qualitative assessments of the Centralised Chronic Medication Dispensing and Distribution (CCMDD) programme, which provides chronic medication, including ART, to over 1 million people in South African communities.

► Our analysis is strengthened by the use of 'theories of practice', which highlights the different components of the social practice of engagement in HIV care, and how these interact and may be influenced by CCMDD.

► Participants in our study were part of a clinical trial at a large urban clinic, meaning our findings may not be generalisable to other settings, such as rural areas.

► Lastly, the CCMDD programme is adapting quickly, and some of the early implementation problems that we report may now have been resolved.

## BACKGROUND

Expanding access to life-saving antiretroviral therapy (ART) for people with HIV has been one of the biggest global health successes of the 21[st] century. Since 2004, the number of people on ART globally has increased more than 10-fold to 23.3 million, while HIV related mortality has dropped from 1.7 million annual deaths to 0.8 million.[1] ART is now recommended for all people living with HIV, meaning that a further 14.6 million people

still require treatment, mainly in low- and middle-income countries.[1] Therefore, ART programmes in these settings will need to continue expanding, while also providing treatment through efficient, convenient services that encourage life-long engagement in HIV care.

Several new models of 'differentiated ART delivery' have been developed, including providing ART through community adherence clubs (where clients meet in a community group to collect ART), delivering ART to decentralised pickup points and increasing the intervals between ART collection dates using multi-month prescribing.[2] These services aim to provide streamlined, efficient treatment in a client-centred manner for people who are doing well on ART, thereby allowing more resources to be directed towards initiating people on treatment and to managing those who are less well.[3 4]

South Africa currently runs the largest differentiated ART delivery programme in the world as part of the Centralised Chronic Medication Dispensing and Distribution (CCMDD) programme.[5] The CCMDD programme allows people with and without HIV to collect prepacked chronic medications (eg, ART, antihypertensives, diabetes treatment, lipid-lowering medication,) in the community, instead of having treatment dispensed at clinics. People living with HIV can be referred into CCMDD by their ART clinic if they are non-pregnant, stable on ART and have had two consecutive suppressed HIV viral loads, at least 6 months apart. In CCMDD, clients collect their treatment every 2 months at local pickup points, including private pharmacies and community-based organisations.[6] Every 6 months, they return to the clinic for review and re-referral into CCMDD. An annual viral load is also performed at the clinic assessing whether they remain virally suppressed. If clients in CCMDD are unwell, they can return to their clinic at any time for medical attention. CCMDD is a public-private partnership; patients are referred by healthcare workers in government sector clinics, while a private company is responsible for centrally dispensing medication and delivering it to community pickup points.[5 7] CCMDD was started in 2014 and provides treatment to over 1.7 million people.[8] However, despite its size, there is little published data evaluating how this programme influences engagement in HIV care.

Several different frameworks and theories have been proposed to evaluate factors influencing engagement in HIV care. Skovdal *et al*[9] propose a theoretical framework around engagement in HIV care as a social practice (figure 1). 'Theories of practice' conceptualise patterns of behaviour as social practices which depend on *materialities* (eg, clinical accessibility), *competencies* (eg, knowing how to access services and/or maintain healthy lifestyle), *meanings* (eg, HIV-associated stigma, the normalisation of HIV as a common experience) and *other life practices* (eg, employment).[9] These different components dynamically interact to influence the 'practice' of engagement in HIV care. Here, we aim to use this framework to explore engagement in clinic-based HIV services, and to analyse how CCMDD may influence the social practice of engagement in HIV care.

## METHODS
### Study design and setting
We undertook semi-structured in-depth interviews and focus group discussions with clients, and interviews with healthcare workers, who were part of the Simplified TREAtment and Monitoring (STREAM) study, a clinical trial of point-of-care HIV viral load testing and task shifting to enrolled nurses.[10 11] We previously reported qualitative findings from this study regarding the acceptability of point-of-care viral load testing and enrolled nurse care.[12] Here, we focus on findings related to CCMDD.

Eligible participants in STREAM were adults aged ≥18 years who were stable on first-line ART for 6 months after initiation. At enrolment, participants were randomised 1:1 to receive point-of-care viral load testing and potential task shifting to enrolled nurses, versus standard laboratory viral load testing and professional nurse care. After 6 months in the study (1 year on ART) participants in both arms who had a suppressed viral load and met other eligibility criteria (clinically stable, non-pregnant, CD4 count >200 cells/mm$^3$)[6] were offered referral into CCMDD.

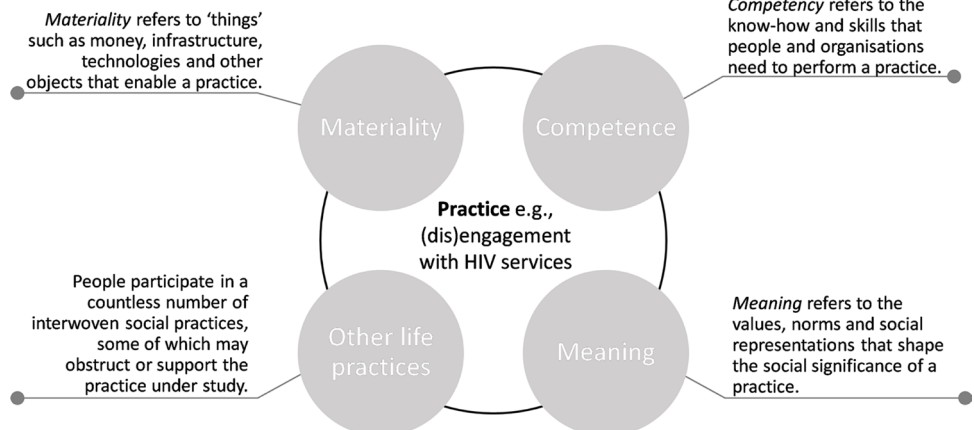

**Figure 1** Summary of theories of practice framework, reproduced with permission from Skovdal *et al*, 2017.[7 9]

**Table 1** Client in-depth interview sampling frame

| Viral load result | Care provider | Point-of-care arm | Standard-of-care arm | Total target |
|---|---|---|---|---|
| | | Target | Target | Total target |
| <1000 copies/ml | CCMDD and/or enrolled nurse | 5 | 5 | 10 |
| | Professional nurse | 5 | 5 | 10 |
| >1000 copies/ml | Professional nurse | 5 | 5 | 10 |
| Total target | | 15 | 15 | 30 |

CCMDD, Centralised Chronic Medication Dispensing and Distribution programme.

Once accepted into CCMDD, participants were able to collect treatment every 2 months at private pharmacies and community-based organisations. The CCMDD system would automatically send SMS texts to clients' mobile phones to remind them of their ART collection dates. After 6 months they were reviewed at the clinic by a nurse. The STREAM study took place at the Prince Cyril Zulu Clinic and the adjacent Centre for the AIDS Programme of Research in South Africa (CAPRISA) eThekwini Clinical Research Site, which are situated next to the main transport hub in central Durban, South Africa. Prince Cyril Zulu Clinic serves a large, urban population and was one of the first clinics to implement CCMDD in Durban, as part of a pilot implementation programme.

### Participants

We used a purposive sampling frame to select a range of patients including those who received care in CCMDD, who had suppressed and unsuppressed viral loads at any point in the study and who received point-of-care testing versus standard laboratory testing in the STREAM trial (table 1). We approached potential participants in person or by telephone after they had exited the STREAM study, explained reasons for performing the research and obtained written informed consent for their participation.

### Data collection

Trained research assistants (LM (male), HS (female)) performed interviews and focus groups in isiZulu or English in a private room in the clinic, using semi-structured topic guides (online supplementary material). LM was not previously known to participants. HS was known to patients as she was involved in recruitment and scheduling in STREAM. Neither LM or HS had any involvement in CCMDD or clinical patient care. Focus groups were broadly organised by STREAM study arm, and by attendance in CCMDD. We audio recorded interviews and focus groups, and took accompanying field notes where appropriate. Recordings and notes were anonymised, transcribed and translated from isiZulu to English where necessary. Participants did not review transcripts or the analysis.

### Data analysis and theoretical framework

We initially performed a deductive thematic analysis[13] using a framework based on predefined themes informed by theories of practice.[9] During analysis, we inductively identified new themes and codes that were derived from the data and integrated them into the predefined themes in the coding frame (online supplementary table S1). Data for this analysis was predominantly coded by JD, with assistance by LM, using NVivo 12.0 (QSR International, Melbourne, Australia).

### Patient and public involvement

Patients and the public were not involved in the design, recruitment or conduct of the study, beyond being enrolled as participants. Results will be shared with the CAPRISA Community Advisory Board as part of dissemination activities.

### RESULTS

Between March 2018 and August 2018 we undertook 25 interviews and four focus groups (5 to 10 participants) with a total of 55 clients, and eight interviews with healthcare workers. One client was both interviewed and also in a focus group. Among clients, median age was 31 years, 56.4% were female, 52.7% were employed and 58.2% had collected ART in CCMDD (table 2). Among healthcare workers the median age was 39, 75.0% were women, and all were nurses (table 2). Interviews lasted 30 to 60 min and focus groups 60 to 80 min. Below, we present our findings, with supporting quotes, regarding the challenges to engagement in clinic-based services, and the influence of CCMDD on the social practice of engagement in HIV care. We use themes based on the four constructs of *materialities*, *competencies*, *meanings* and *other life practices* from theories of practice,[9] as well as the theme of problems with CCMDD implementation which we identified in the analysis (online supplementary table S1).

### Materialities

Clients and healthcare workers described several ways in which the *materialities* of clinic-based HIV services presented challenges to ART provision, with overcrowded

 

**Table 2** Participants demographic and clinical characteristics

**Clients**

| | | Frequency (n=55) | % |
|---|---|---|---|
| Age (years) | Median (IQR) | 31 (27 to 37) | |
| Gender | Male | 24 | 43.6 |
| | Female | 31 | 56.4 |
| Ethnicity | Black African | 55 | 100.0 |
| Educational level | None or primary school | 2 | 3.6 |
| | Did not pass secondary school | 14 | 25.5 |
| | Passed secondary school | 27 | 49.1 |
| | Some tertiary education | 12 | 21.8 |
| Employed | Yes | 29 | 52.7 |
| | No | 26 | 47.3 |
| Income in ZAR per month (approximate USD) | <1000 (70) | 23 | 41.8 |
| | 1000 to 4000 (70 to 280) | 20 | 36.4 |
| | >4000 (280) | 12 | 21.8 |
| Has a regular/stable partner | Yes | 42 | 76.4 |
| | No | 13 | 23.6 |
| Number of children | None | 10 | 18.1 |
| | One or more | 45 | 81.8 |
| Disclosed HIV status to anyone | Yes | 52 | 94.6 |
| | No | 3 | 5.5 |
| Travel time to clinic >1 hour | Yes | 7 | 12.7 |
| | No | 48 | 87.2 |
| Distance travelled to clinic ≤5 kilometres | Yes | 44 | 80.0 |
| | No | 11 | 20.0 |
| Primary method of travel to clinic | Walking | 3 | 5.5 |
| | Public transport | 49 | 89.1 |
| | Private transport | 3 | 5.5 |
| VL <1000 copies/ml | Yes | 49 | 10.9 |
| | No | 6 | 89.1 |
| Collected ART in CCMDD | Yes | 32 | 58.2 |
| | No | 23 | 41.8 |
| **Healthcare providers** | | | |
| Age (years) | Median (IQR) | 39 (36 to 42) | |
| Gender | Male | 2 | 25.0 |
| | Female | 6 | 75.0 |
| Profession | Professional nurse | 4 | 50.0 |
| | Enrolled nurse | 4 | 50.0 |

.ART, antiretroviral therapy; CCMDD, Centralised Chronic Medication Dispensing and Distribution; USD, United Stated Dollar; VL, viral load; ZAR, South African Rand.

primary care clinics and long waiting times deterring people from collecting treatment.

"Sometimes I would end up not even collecting them (ART), because of the thought of the queue." Focus group 2, Participant 9, Female

"The clinic gets very full, you will wait there the whole day." Focus group 3, Participant 3, Male

"When you are coming to the clinic, you must cancel all the plans you have and tell yourself that you will spend the rest of the day there. Because chances are you will finish late in the afternoon." Focus group 2, Participant 1, Male

In contrast, many clients and healthcare workers described how CCMDD introduced changes in the *material circumstances* of ART delivery, with faster ART collection and more convenient opening hours.

"I think collecting at (the private pharmacy) is quicker than here. When I came back here at the clinic, I

waited for a good 2 hours." Focus group 1, Participant 7, Male

"They (patients) like CCMDD, (because) when they go to the pickup points there's no queues like here in the clinic. They come in, they go straight to the pharmacies then they get their pack." Staff interview 4

"Collecting… (at the community-based organisation) saves a lot of time, you don't spend 5 minutes. It doesn't get full, you can come anytime if it's still open. You just get in there, take your medication and leave." Focus group 3, Participant 5, Female

These material differences between clinic-based ART provision and CCMDD, impacted and interacted with the other components of the social practice of engagement in HIV care.

### Competencies

Clients described how time pressures and rushed appointments in clinics prevented communication with healthcare workers, which stopped them from gaining *competency* in understanding and managing their health and HIV treatment.

"They would just tell you your bloods are fine and you leave… We don't ask questions, we want to finish and allow other people to come in… there are a lot of patients (waiting) outside who also want to come in and be assisted." Client interview 3, Male

"Maybe you weren't feeling well 2 days ago and you want to tell the nurse about it, sometimes you can't… because you are looking at the situation the nurse is under. You see that she has no time for you and she will not tolerate you telling her other things." Focus group 3, Participant 4, Female

Some clients also described being treated badly by clinical staff, further limiting their ability to become *competent* at taking medication.

"Sometimes you speak with (the nurse), and she gives you medication, (but) at the same time she's shouting at you. You can't do anything as a patient, you end up not knowing how you should take the medication." Focus group 3, Participant 4, Female

Two healthcare workers expressed concerns that clients who become unwell in CCMDD could then have difficulty accessing clinical care. This could be overcome by communicating well to ensure that clients understood that they could return to the clinic at any time.

"What I don't like about CCMDD is that… you can say the patient is stable, but tomorrow is another story. So if such patients go to CCMDD… and something new develops… they (CCMDD staff) tell you that you must go back to your facility (by which time) … it will be late and the damage is done." Staff interview 7

"I don't think there's anything bad (about CCMDD) as such. But the thing is, if a patient becomes unstable,

then there's nobody to care for them because at (the pickup point) they just issue the medication… the patient must be educated to return to us if they are not feeling well." Staff interview 5

Most clients seemed to understand that quicker ART collection through CCMDD came at the cost of receiving less clinical oversight, but understood that they could return to the clinic if they were unwell or needed additional care.

"It's quicker to just collect (at the community-based organisation), because you know how you are taking your treatment… because when all these things (measuring vital signs, seeing a nurse) are done, you will be delayed." Focus group 2, Participant 8, Female

"…but at (the private pharmacy) you take your medication and leave, and it's good. When you have a problem, you come to the clinic." Focus group 2, Participant 9, Female

However, one client refused referral into CCMDD because she liked having vital signs monitored and the opportunity to talk to nurses in clinic, while another client in CCMDD felt that it could be improved by a quick physical check-up in the community.

"They once said we can now get them (ART) outside… I don't want that, I am fine here (at the clinic)… You weigh, check blood pressure and then go to the nurse to collect medication and she will look at your results and tell you everything is going well, or you are lacking here and you are lacking here." Client interview 16, Female

"I was relieved (that I was in CCMDD) because things were going to be quicker now, because this thing of staying all day at the clinic is not nice. But then the fact that you don't get checked up is not good. Otherwise, it's fine because it (the community-based organisation) is quick. If your vital signs were done as well, it would have been better." Client interview 5, Female

Clients not in CCMDD developed other *competencies* to overcome the multiple challenges to engagement in clinic-based care. These included sending relatives or friends to collect treatment, sharing ART, arriving at clinic very early in the morning to miss queues and extreme patience because of the life-saving benefits and positive *meanings* associated with receiving ART.

"When it's my appointment date I wake up very early to get to the clinic… I will be here for the whole day… because there are a lot of people… You must wait that long and sometimes you get so tired and you wish you can just leave. What can we say? We must wait because it's our lives." Client interview 12, Female

While successfully collecting ART in CCMDD required development of new *competencies*, some clients' previous experiences of buying from pharmacies was helpful.

"Ok, so (the first time I collected from the private pharmacy) I got in there and looked around to see where I'm supposed to go, because I'm used to just buying and then leaving. Ok, I saw that I need to go (to) the dispensary because I am coming to collect and I didn't have a problem I did everything well and I found a good person who was full of jokes." Client interview 18, Female

### Meanings

The physical organisation of clinical services sometimes threatened clients' confidentiality, potentially revealing their HIV status, which for them *meant* potential exposure to HIV-related stigma.

"There are two polyclinics, there's an old one and the new one… (HIV) positive people go to the 'old poly', so when a person says, "He is an old poly" (it means) they saw you at the 'old poly'. They know when you go to the 'old poly' that you are coming for ART." Focus group 1, Participant 1, Male

In order to manage the risk of HIV-related stigma in clinics, clients developed *competencies* to avoid disclosing their HIV status to people that they knew. Some reported pretending to collect treatment for someone else, or choosing to collect their treatment at a clinic far from home.

"(There is) another thing that makes us collect medication in other areas. We do have clinics in our local communities, (but) we are afraid to collect medication in those clinics because it is not safe… we are in one room with other patients and maybe we are only separated by a wall, so when I enter that door, people will see that I am going to collect my 'qo' (ART)." Focus group 3, Participant 7, Male

In contrast, several staff felt that CCMDD helped to change the *meanings* associated with HIV treatment by normalising HIV services.

"Being on ART, it disrupted their (clients) normal lives, so with CCMDD people go back to their normal life routine. You don't have to feel that 'I am sick'… you don't have to be burdened with getting treatment." Staff interview 1

Specifically, collecting ART in private pharmacies, where clients could be getting non-HIV related medication, helped with maintaining confidentiality.

"(In the private pharmacy) they wrap the medication well and pack it well. They don't just give you unwrapped medication where everyone can see. They wrap them and put them in a plastic bag… it looks like any medication you buy from (the pharmacy)." Client interview 12, Female

However, as we found in clinics, stigma still affected the *meanings* of engagement with CCMDD. Some clients who collected from private pharmacies felt that they were treated differently, because they were not paying customers.

"They (the private pharmacy staff) were paying attention to the customers who were coming to buy. Because we don't come with money, we only come with our (CCMDD) cards, they weren't paying attention to us." Client interview 8, Female

In some pharmacies there was a separate queue for CCMDD clients, which they felt could expose their HIV-positive status to others, despite people without HIV also collecting other chronic medication through CCMDD.

"Where I collect, there is our (CCMDD) queue and then there is a queue for everyone else. That is why I say sometimes they would say 'these people are coming to collect their thing (ART)'." Focus group 1, Participant 1, Male

Furthermore, integrating HIV treatment with everyday life felt threatening to some clients. One declined referral to CCMDD because he was worried about revealing his HIV status and experiencing stigma.

Interviewer: "Okay, why don't you like going there (private pharmacy in CCMDD)?"

Participant: "Uhm… because this thing (collecting ART) is confidential. Imagine, since I'm working at the mall, how many people who live by my house come there?… Imagine a female who stays by my place and sees me when I go to this. This is a secret… So, I would prefer to collect them here (at the clinic), because I don't want to feel embarrassed." Client interview 3, Male

In general, clients understood that CCMDD was for stable patients, and associated being referred into CCMDD as *meaning* that they had been taking ART well. Some used this to motivate their own adherence.

"No one wants to go back to the clinic anymore, and I think that is why we are taking our medication so well… When the nurse showed me my results, I was relieved that I will be going back (to CCMDD)." Focus group 2, Participant 9, Female

### Other life practices

Many clients found that long waiting times in clinics and inflexible clinic opening hours (weekdays between 0700 to 1600 hours) clashed with employment opportunities and family responsibilities:

"Yes, but if I could be able to collect it on weekends… During the week I sometimes have temporary jobs somewhere, so I wish I could collect on weekends." Client interview 18, Female

"I did go to the clinic and the line was massive. So, I couldn't collect (ART), because I had to go back to work." Focus group 2, Participant 4, Male

"…because the time here (at clinic) clashes with my work time… When you come here (to the clinic) you do it for yourself and when you are at work you do it for the family, you see. You must choose sometimes." Client interview 24, Male

Quicker services in CCMDD and more flexible opening times meant that collecting ART was less disruptive for *other life practices.*

"Everything (at the community-based organisation) is perfect and fast. If… I'm supposed to start work at half past 8 I can make it, because at 8 o'clock I will be there and by 10 past or quarter past I'm done." Client interview 14, Male

"(In CCMDD) the patients… are independent, they can go on a Saturday or after hours to pick up their medication, they don't have to take time off work to come to us (at the clinic)." Staff interview 5

However, some pharmacies seemed to be placing restrictions on when ART could be collected, despite being open for other services. This inflexibility meant that some employed clients still had difficulties collecting their ART in CCMDD.

"The problem with (this private pharmacy) is that they don't issue treatment on weekends. So, since we are working, we don't have time because if I didn't come to work they will deduct a certain amount from my salary. When you go there in the afternoon, they tell you that they don't issue medication at the certain times. Which means I will not be getting my treatment whereas it is very important to me." Focus group 1 Participant 9, Female

**Challenges with implementation of CCMDD:** Despite the general benefits of CCMDD, a few staff and clients also reported occasional problems with the implementation of the programme (table 3). These included delays in receiving reminder SMS, the client's ART not being found at the pickup point and inflexible pickup dates at some pharmacies, with pickup points sending ART back to

**Table 3** Challenges with implementation of CCMDD

| Issue | Quotes |
| --- | --- |
| **Barriers to registering patients in CCMDD** | |
| Need for ID card | "If a patient wants to be registered on CCMDD, strictly that patient must have a South African ID (or foreign passport) and every time the patient is going to collect the medication, they have to positively identify themselves using their ID. Something that we are not doing in the clinics. Some patients are not happy with that, ID's do get lost from time to time and then once the ID is not there, then they cannot give you that package at the pickup point, then the patient would be forced to come back to the clinic… (and) then they get de-registered in the process." Staff interview 4* |
| Problems with electronic prescription system | "The online registration system crashes quite often so there's days where the nurses can't get online, they just have to register manually. It creates a problem because online registration is a lot quicker, so on the days when they have to do manual then they are backed up for quite a while, and then it just makes the work a lot more." Staff interview 6 |
| TB prophylaxis | "There's a category of patients that can be stable on ARV's but still don't qualify for CCMDD. A typical point in example, if at any given point you decide to put this patient on IPT, which is Isoniazid Preventive Therapy for TB, they don't take that patient. You see, for the patient to qualify for CCMDD they must be stable, but now CCMDD will not take a patient that is on IPT. So we have to keep that patient up until they finish the course of IPT before you can enrol them on CCMDD." Staff interview 4<br>"They said because I am still taking the ones for protecting me against TB and I will finish them in August… after that I will hear from them (about referral to CCMDD)" Client interview 21, Male. |
| **Problems with CCMDD implementation** | |
| SMS | "I will receive the text message the following day in the afternoon, after I collected today. I never receive the text message (on time). I receive it after." Focus group 1 Participant 3, Female |
| ART not being found at the pickup point | "I once received an SMS and I went there (to the private pharmacy) the very same day I received the SMS, and I got there and they said my name is not there (on the list of people with ART to collect). I then came here (to the clinic), and I got my treatment here. Fortunately, I had time on that day to end up coming here." Focus group 2 Participant 4, Male |
| Inflexible pickup dates | "The service providers are very strict when it comes to appointment dates, very strict. You get given a date and then you are expected to collect your parcel within 48 hours. After those 2 days then the medication will be sent back… and they get de-registered as a result. But here in the clinic they can miss the date by a day or two or three or four or five, it's not that much of an issue." Staff interview 4<br>"It was my fault because I came after 2 days and I found that my treatment has been sent back (to the depot)." Focus group 1 Participant 6, Male |
| Restricted ART pickup times | "They (private pharmacy staff) don't even take into consideration that I am coming from work… They are still at work and their job is to give me what I came here for (ART), but they tell me that they have closed and there is a queue." Focus group 1, Participant 1, Male |
| Pickup points reaching capacity | "Most of the pickup points were full, you know. They were full and some of the participants collect their medication far from where they stay, far from where they work you know. Jah, I think those were the disadvantages." Staff interview 2 |

ART, antiretroviral therapy; CCMDD, Centralised Chronic Medication Dispensing and Distribution programme; TB, tuberculosis.

the central depot, if clients were a few days late (table 3). At the time of the interviews, several pickup points had reached capacity and were no longer accepting referrals, meaning that some clients had no pickup points near to their homes. Staff also reported some barriers to registering people into CCMDD. Referral criteria require a valid identification card or passport, which excluded some clients. Furthermore, isoniazid prophylaxis therapy, a daily tablet to prevent tuberculosis, is not available in CCMDD, meaning clients could not be referred until they had completed the recommended 12-month prophylactic course. Lastly, the electronic prescription system for CCMDD was occasionally offline, which could make CCMDD registration more time consuming (table 3).

## DISCUSSION

In this study, clients and healthcare workers identified several problems with collecting ART from clinics in Durban, South Africa. Using the themes of materialities, competencies, meanings and other life practices based on 'theories of practice',[9] we showed how these problems interact with, and impact on, engagement in HIV care. We present one of the first qualitative assessments of the largest differentiated ART delivery programme in the world and show how it overcame some of these problems, helped to normalise HIV treatment and reduced the impact of collecting ART on other social practices in the clients lives. However, several challenges and inconsistencies remain in the implementation of CCMDD.

### Materialities

We found several material barriers to engagement in clinic-based services, which are similar to findings from other studies. In a qualitative systematic review of 59 studies regarding long-term ART adherence and engagement in care in Africa, poor clinical services were identified as a major theme, with long waiting times, inflexible services and rigid and demanding policies all leading clients to disengage from care.[14] In our study, CCMDD changed some of the 'materialities' of ART provision, leading to clear practical benefits for clients. Similar findings have been reported in qualitative assessments of community adherence clubs in Mozambique, Kenya, Malawi, Zimbabwe and South Africa, which found reductions in clinical visits, waiting times and transport costs for clients.[15–19]

### Competencies

The material challenges to attending clinics interacted with other components of the social practice of engagement in HIV services. For example, in our study and in another study from South Africa,[20] rushed clinical appointments impeded communication with nurses, meaning there was no time for healthcare workers to equip patients with the competencies required to engage with HIV services and learn about taking ART. Accessing ART through the CCMDD programme also required

clients to develop new skills and competencies. When the CCMDD system worked well, clients generally found it easy to navigate, and understood that they could return to their normal clinic if they needed additional care. However, a few healthcare workers expressed concerns that CCMDD could delay clients from accessing appropriate medical attention if they became unwell, and highlighted the need for adequate counselling of clients to ensure appropriate healthcare seeking. These findings are similar to a study of community adherence groups in Zimbabwe, where healthcare workers were worried about delayed access to care, while clients were happy to spend less time in clinics, and felt 'empowered to visit the clinic whenever they needed'.[19]

### Meanings

We found that healthcare workers in clinics were often overburdened, sometimes disrespectful, and did not always maintain confidentiality, as has been seen in many other studies.[14] Clinics that separated HIV services from other care also threatened clients' confidentiality, therefore exposing clients to stigma, which has been widely reported as impacting negatively on engagement in HIV care.[21 22] CCMDD also impacted on the meanings of engagement in HIV care. Attending private pharmacies was less burdensome, less obviously associated with being HIV positive, and more easily integrated with everyday life, thereby helping to normalise ART collection. In Kenya, integrated adherence clubs for non-communicable diseases and HIV were reported to reduce stigma, by normalising HIV to be like other chronic conditions.[16] However, we found multiple reports of preferential treatment by private pharmacy staff towards paying customers, suggesting some ongoing discrimination towards public sector CCMDD clients. Some clients also remained concerned that their HIV status could be revealed when collecting at community pickup points, where they could be more likely to see neighbours or people they knew. In studies of community adherence clubs, clients were mainly concerned about revealing their HIV status to other group members, and therefore created strict rules around maintaining group confidentiality.[15–18] The risk of other adherence club members breaching confidentiality was offset by increased peer support from within the group.[15 17 19 20] In contrast, clients in CCMDD did not report benefits of peer support, but also did not have to negotiate group dynamics when collecting ART, as they tended to collect individually at pickup points. Lastly, referral into CCMDD provided a new positive meaning to ART collection, and was seen as a reward and incentive to take ART well. Similarly, adherence clubs clients felt like staff were trusting them to manage their own HIV, and saw being sent back to the clinic as a punishment or failure to adhere to treatment.[20]

### Other life practices

Several studies have described how the time spent travelling to clinics and waiting for ART can impact on clients'

ability to seek or attend employment, and meet family responsibilities.[15 17 19] We found that a major benefit of CCMDD was reduced disruption to employment, although some working clients encountered problems with restricted collection times in a few private pharmacies. Similarly, in differentiated care programmes in Malawi, Kenya and Zimbabwe, clients had more time to attend their businesses and farming,[17 19] and avoided taking days off work.[16 19]

### Challenges with CCMDD implementation

As with other differentiated ART delivery programmes,[15–18] there were several reports of implementation problems that clients had to negotiate, such as delayed SMS reminders, ART not being available at the pickup point and a few pharmacies placing restrictions on ART pickup times. Furthermore, some of the eligibility criteria for CCMDD seemed to contradict other stated aims of the healthcare system. For example, the need for an identification card could exclude the most vulnerable clients, while not providing tuberculosis prophylaxis within CCMDD could hamper tuberculosis control measures.[23]

### Strengths and limitations

This is one of the first qualitative studies to assess differentiated ART delivery within the CCMDD programme. Our analysis is strengthened by the use of 'theories of practice' which highlighted the relationship between the materialities of ART provision, and how changes in these impacted on engagement in HIV care as well as clients' broader lives. Our interviewers were not involved in CCMDD or clinical patient care, which reduces the potential for social desirability bias. Limitations of our study include the use of a single urban clinic, and the use of clinical trial participants, meaning our findings may not be generalisable to other settings, such as rural areas. While our purposive sampling aimed to include a range of clients, clients who felt more positive towards CCMDD may have been more likely to participate. Lastly, the CCMDD programme is adapting quickly, and some of the early implementation problems that we report (eg, pickup points reaching capacity) may now have been resolved.

### Implications

Our findings support the ongoing roll-out of CCMDD, while highlighting areas where improvements are needed. Better regulation of private pharmacies could ensure consistent implementation of less restrictive ART collection times, and allow development of strategies to better protect confidentiality and prevent CCMDD clients from receiving inferior care. Increasing the flexibility of the ART collection date window could also increase clients' autonomy to collect on a day that suits them in CCMDD. Steps to reduce HIV-related stigma at a broader societal level are needed, as fear of disclosure remained a barrier to engagement in both clinic-based services and CCMDD. Further research is needed to establish the impact of CCMDD on clinical outcomes such as viral suppression

and retention in care, and whether it is cost-effective. Of note, cost-effectiveness studies should attempt to capture the wider non-healthcare system costs and benefits for employed people, given that CCMDD may be particularly beneficial to them. Lastly, we also identify several areas that could be improved in clinic-based HIV services, including less restrictive opening times, more respectful treatment of clients and better confidentiality.

## CONCLUSION

In conclusion, we highlight how the material changes to ART provision in CCMDD facilitated engagement with HIV services, reduced disruption in clients' everyday lives and helped to normalise HIV care. Improvements to the implementation of CCMDD could allow even greater benefits, and contribute towards achieving universal ART through more client-centred healthcare services.

**Author affiliations**
[1]Nuffield Department of Primary Care Health Sciences, University of Oxford, Oxford, UK
[2]Centre for the AIDS Programme of Research in South Africa (CAPRISA), University of KwaZulu-Natal, Durban, South Africa
[3]Gender and Health Research Unit, South African Medical Research Council, Pretoria, South Africa
[4]Centre for Rural Health, School of Nursing and Public Health, University of KwaZulu-Natal, Durban, South Africa
[5]NIHR Health Protection Research Unit in Healthcare Associated Infections and Antimicrobial Resistance, University of Oxford, Oxford, UK
[6]eThekwini Municipality Health Unit, Durban, South Africa
[7]Department of Global Health, Schools of Medicine and Public Health, University of Washington, Seattle, United States
[8]Department of Medicine, School of Medicine, University of Washington, Seattle, United States
[9]Department of Epidemiology, School of Public Health, University of Washington, Seattle, United States
[10]Discipline of Public Health Medicine, School of Nursing and Public Health, University of KwaZulu-Natal, Durban, South Africa

**Acknowledgements** We would like to thank Farzana Osman for descriptive analysis of participants' demographics. The support of the DST-NRF Centre of Excellence for HIV Prevention towards this research is hereby acknowledged. Opinions expressed, and conclusions arrived at, are those of the authors and are not necessarily to be attributed to the DST-NRF Centre of Excellence for HIV Prevention.

**Contributors** JD, AG, HN, PKD and NG conceived and designed the study. LM and HS conducted interviews and focus group discussions, and transcribed and translated the data. JD, NG, HN and AG supervised data collection. JD and LM coded and analysed the data. AG, STC, GH and CB supervised data analysis. JD drafted the manuscript. JD, LM, AG, HS, STC, GH, CB, HN, PKD and NG critically reviewed and edited the manuscript and consented to publication.

**Funding** The STREAM study was funded by the US National Institute for Health (AI124719-01). JD is funded by the Wellcome Trust PhD Programme for Primary Care Clinicians (216421/Z/19/Z). LM was funded by DST-NRF Centre of Excellence for HIV Prevention (UID96354). AG is supported by the South African Medical Research Council. STC was supported by funding from the National Institute for Health Research (NIHR) Health Protection Research Unit in Healthcare Associated Infections and Antimicrobial Resistance at University of Oxford in partnership with Public Health England (HPRU-2012-10041).

**Competing interests** None declared.

**Patient and public involvement** Patients and/or the public were not involved in the design, or conduct, or reporting or dissemination plans of this research.

**Patient consent for publication** Not required.

**Ethics approval** Ethical approval for this study was provided by the Biomedical Research Ethics Committee of the University of KwaZulu-Natal and the Institutional Review Board of the University of Washington.

**Provenance and peer review** Not commissioned; externally peer reviewed.

**Data availability statement** No data are available.

**ORCID iDs**
Jienchi Dorward http://orcid.org/0000-0001-6072-1430
Andrew Gibbs http://orcid.org/0000-0003-2812-5377
Sarah Tonkin-Crine http://orcid.org/0000-0003-4470-1151
Gail Hayward http://orcid.org/0000-0003-0852-627X
Christopher C Butler http://orcid.org/0000-0002-0102-3453
Hope Ngobese http://orcid.org/0000-0002-1730-8904
Paul K Drain http://orcid.org/0000-0002-9569-4097
Nigel Garrett http://orcid.org/0000-0002-4530-234X

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
