## [Reviewer comments · BMJ Open]

ARTICLE DETAILS

TITLE (PROVISIONAL)	Understanding how community antiretroviral delivery influences engagement in HIV care: a qualitative assessment of the Centralised Chronic Medication Dispensing and Distribution programme in South Africa
AUTHORS	Dorward, Jienchi; Msimango, Lindani; Gibbs, Andrew; Shozi, Hlengiwe; Tonkin-Crine, Sarah; Hayward, Gail; Butler, Christopher C.; Ngobese, Hope; Drain, Paul K; Garrett, Nigel

VERSION 1 – REVIEW

REVIEWER	Tom Decroo Institute of Tropical Medicine Antwerp, Belgium
REVIEW RETURNED	14-Nov-2019

GENERAL COMMENTS	Dear authors, With interest I read “Understanding how community antiretroviral delivery programmes influence engagement in HIV care: a qualitative assessment of the Centralised Chronic Medication Dispensing and Distribution programme in South Africa”. Your study is very relevant. In general the manuscript is very well structured and easy to read. Recommendations for major revisions: • Results: at the end of the intro you refer to 4 broad thematics (materialities, competences, meanings, other life practices), part of the framework you used for the deductive approach to data analysis. In the methods you mention that, using a deductive approach, these predefined themes were explored, and that other themes emerged from the data. Therefore I recommend to show the results using the same 4 sections (one for each of these 4 themes) and one (or more than one) section with the additional themes. This would, in my opinion, improve coherence.• In my experience, barriers to community ART delivery implementation include worries about the clinical status of patients, as many patients were found not “stable” enough. Others are worried about timely referral in case of illness when patients receive ART through a community model. I would like to know more about these clinical “themes”, especially as you interviewed patients with an unsuppressed viral load. Data on these themes are needed to inform the debate on community-based ART with those who have a more clinical perspective.• You also interviewed those in clinic-based care, and mention that a few interviewees preferred continuing having a periodic clinical examination (lines 272 - ...). I think it’s not explicitly stated if these patients refused joining the community model, or showed preference for clinic-based refill. If you have the data, please
---

	expand on reasons for preferring clinic-based care or reasons for requests to return to clinic-based care. Minor revision  • Abstract, line 16: “in clinics ...” seems to generalized ... I guess you refer to one service: clinic-based ART refill. • Methods -analysis (line 125 -...): Would you not consider the analytical approach a combination of inductive and deductive analysis? You started from a theoretical framework but were open to identifying new themes?
--	---

REVIEWER	David Barr Joep Lang Institute, USA
REVIEW RETURNED	18-Dec-2019

GENERAL COMMENTS	The study provides a good description of the drug distribution program, why is it considered useful for patients in comparison with drug pick up at a clinic. However, the sample size is quite small and participants come from one part of the country. It would be interesting to know if the opinions of the study participants were different from different geographic locations and/or if there were more diverse opinions if a larger number of people were interviewed. The framework used to structure the interviews is interesting and reasonable, though it is difficult to know all the factors of patient experience that were included in each of the four categories and, therefore, whether these categories were sufficient to encompass the concerns, needs and priorities of the interviewees. That said, this paper provides keen insight into the experience of people picking up their drugs at a pharmacy instead of spending a great deal of time at a clinic to pick up medicines. It's good to see this, but much of it seems like common sense. It is hard to imagine how the results could be other than they are.
---

REVIEWER	Sheree Schwartz Johns Hopkins University
REVIEW RETURNED	05-Jan-2020

GENERAL COMMENTS	BMJ Open 2019-035412 Overall this is a very well written manuscript tackling an important topic, reflections on the implementation of the chronic care dispensing program in South Africa. Although the noted barriers to clinic care were not surprising and replicate findings from other studies, the section focusing on Centralised Chronic Medication Dispensing and Distribution (CCMDD) implementation was interesting and provided new insights. Clearer highlighting in the abstract regarding findings around CCMDD implementation heterogeneity would guide readers to what I found to be the most compelling elements of the article. Minor comments: Introduction: Line 69: Please clarify if anti-hypertensives are the only non-HIV drug dispensed or provide additional examples/list all (if reasonable) Methods: Line 109: Based on Table 1, selection of participants including virally suppressed and non-virally suppressed appears to be
--

	based on baseline or 6-month viral loads, not viral loads at the end of the study. Please clarify in line 109. Line 112: Please clarify if informed consent was written or verbal Data Collection: How were focus groups organized or stratified (by sex/gender, by CCMDD vs. non-CCMDD, etc) over was everyone interviewed together independent of the reasons that they were purposively sampled? Results: Were there any differences by age or gender? The sample is well balanced at least in terms of gender, but it is not clear if there were any differences between men and women (including the gender of the individuals providing the quotes) which would be interesting to understand. Most of the CCMDD engagement info focused on those attending pharmacies. Were there any insights from those in non-pharmacy settings? Because patients had to present an ID, did this mean that they could no longer send family/friends? If so, how was this perceived? Discussion: Line 314: assessment should be assessments
--	--

VERSION 1 – AUTHOR RESPONSE

Reviewer: 1

Comment 1: Dear authors, With interest I read “Understanding how community antiretroviral delivery programmes influence engagement in HIV care: a qualitative assessment of the Centralised Chronic Medication Dispensing and Distribution programme in South Africa”. Your study is very relevant. In general the manuscript is very well structured and easy to read.

Response: We thank the reviewer for their comments and suggestions which have improved the manuscript.

Major comment

Comment 2: Results: at the end of the intro you refer to 4 broad thematics (materialities, competences, meanings, other life practices), part of the framework you used for the deductive approach to data analysis. In the methods you mention that, using a deductive approach, these predefined themes were explored, and that other themes emerged from the data. Therefore I recommend to show the results using the same 4 sections (one for each of these 4 themes) and one (or more than one) section with the additional themes. This would, in my opinion, improve coherence.

Response: As suggested, we have now completely restructured the results section, as well as parts of the abstract and discussion, using the 4 broad themes from the theory of practice framework, with an additional section on CCMDD implementation.

Lines 11-40 (abstract), 160-430 (results) 434-513 (discussion): Please see the manuscript for the restructured sections

Comment 3: In my experience, barriers to community ART delivery implementation include worries about the clinical status of patients, as many patients were found not “stable” enough. Others are worried about timely referral in case of illness when patients receive ART through a community model. I would like to know more about these clinical “themes”, especially as you interviewed patients

with an unsuppressed viral load. Data on these themes are needed to inform the debate on community-based ART with those who have a more clinical perspective.

Response: Thank you for noting this. While this was not a major theme in our findings, it was mentioned by a few healthcare workers. We have now added the following to the results, and also to the discussion:

Lines 234-253: ***“Two healthcare workers expressed concerns that clients who become unwell in CCMDD could then have difficulty accessing clinical care. This could be overcome by communicating well to ensure that clients understood that they could return to the clinic at any time.*”**

“What I don’t like about CCMDD is that... you can say the patient is stable but tomorrow is another story. So if such patients go to CCMDD... and something new develops... they [CCMDD staff] tell you that you must go back to your facility [by which time] ... it will be late and the damage is done.” Staff interview 7

“I don’t think there’s anything bad [about CCMDD] as such. But the thing is, if a patient becomes unstable, then there’s nobody to care for them because at [the pickup point] they just issue the medication... the patient must be educated to return to us if they are not feeling well.” Staff interview 5

Most clients seemed to understand that quicker ART collection came at the cost of receiving less clinical oversight at the pick-up point, but understood that they could return to the clinic if they were unwell or needed additional care.

“It’s quicker to just collect [at the community based organization], because you know how you are taking your treatment... because when all these things [measuring vital signs, seeing a nurse] are done, you will be delayed.” Focus group 2, Participant 8, Female

“...but at [the private pharmacy] you take your medication and leave, and it’s good. When you have a problem, you come to the clinic.” Focus group 2, Participant 9, Female

And

Lines 462-471: ***“Accessing ART through the CCMDD programme also required clients to develop new skills and competencies. When the system worked well, clients generally found it easy to navigate, and understood that they could return to their normal clinic if they needed additional care. However, a few healthcare workers expressed concerns that CCMDD could delay appropriate medical attention for clients who became unwell, and highlighted the need for adequate counselling of clients to ensure appropriate healthcare seeking. These findings are similar to a study of community adherence groups in Zimbabwe, where healthcare workers were worried about delayed access to care, while clients were happy to spend less time in clinics, and felt “empowered to visit the clinic whenever they needed”. [17]”***

Comment 4: You also interviewed those in clinic-based care, and mention that a few interviewees preferred continuing having a periodic clinical examination (lines 272 - ...). I think it’s not explicitly stated if these patients refused joining the community model, or showed preference for clinic-based refill. If you have the data, please expand on reasons for preferring clinic-based care or reasons for requests to return to clinic-based care.

Response: As suggested, we have expanded on the quotes from interviewees who requested clinical examination

Lines 245-265: “**Most clients seemed to understand** that quicker ART collection came at the cost of receiving less clinical oversight at the pick-up point, but understood that they could return to the clinic if they were unwell or needed additional care.

“It’s quicker to just collect, because you know how you are taking your treatment... because when all these things [measuring vital signs, seeing a nurse] are done, you will be delayed.”
Focus group 2, Participant 8

“...but at [the private pharmacy] you take your medication and leave, and it’s good. When you have a problem, you come to the clinic.” Focus group 2, Participant 9

However, **one client refused referral into CCMDD because she liked having vital signs monitored and the opportunity to talk to nurses in clinic, while another client in CCMDD felt that it could be improved by a quick physical check-up in the community.**

“They once said we can now get them [ART] outside... I don’t want that, I am fine here [at the clinic]... You weigh, check blood pressure and then go to the nurse to collect medication and she will look at your results and tell you everything is going well, or you are lacking here and you are lacking here.” Client interview 16

“**I was relieved [that I was in CCMDD] because things were going to be quicker now, because this thing of staying all day at the clinic is not nice. But then the fact that you don’t get checked up is not good. Otherwise, it’s fine because it [the community-based organization] is quick. If your vital signs were done as well, it would have been better.**”
Client interview 5, Female”

Minor Comments

Comment 5: Abstract, line 16: “in clinics ...” seems too generalized ... I guess you refer to one service: clinic-based ART refill.

Response: As suggested we clarify that we are referring to ART refill as part of HIV care.

Lines 19-20: “**For standard clinic-based ART provision...**”

Comment 6: Methods -analysis (line 125 -...) : Would you not consider the analytical approach a combination of inductive and deductive analysis? You started from a theoretical framework but were open to identifying new themes?

Response: We agree with the reviewer that this was more of a combination of deductive and inductive analysis and have highlighted this below:

Lines 143-146: “We **initially** performed a deductive thematic analysis[10] using a framework based on predefined themes informed by theories of practice.[7] **During analysis**, we **inductively** identified new themes and codes that were derived from the data and integrated them into the pre-defined themes in the coding frame (Table S1).”

Reviewer: 2

Comment 7: The study provides a good description of the drug distribution program, why is it considered useful for patients in comparison with drug pick up at a clinic. However, the sample size is

quite small and participants come from one part of the country. It would be interesting to know if the opinions of the study participants were different from different geographic locations and/or if there were more diverse opinions if a larger number of people were interviewed.

Response: We thank the reviewer for their comments and agree that a more geographically diverse sample would be interesting. We discuss this limitation in the article summary and the discussion:

Lines 49-51: "*Participants in our study were part of a clinical trial at a large urban clinic, meaning our findings may not be generalisable to other settings, such as rural areas.*"

Lines 533-535: "*Limitations of our study include the use of a single urban clinic, and the use of clinical trial participants, meaning our findings may not be generalisable to other settings, such as rural areas.*"

Comment 8: The framework used to structure the interviews is interesting and reasonable, though it is difficult to know all the factors of patient experience that were included in each of the four categories and, therefore, whether these categories were sufficient to encompass the concerns, needs and priorities of the interviewees.

Response: Thank you for this comment which was also touched upon by Reviewer 1. We now highlight in the methods section that while we based the initial analysis on the theories of practice framework, we identified new themes were not encompassed by this framework and included them in the analysis.

Lines 143-146: "*We **initially** performed a deductive thematic analysis[10] using a framework based on predefined themes informed by theories of practice.[7] **During analysis**, we **inductively** identified new themes and codes that were derived from the data and integrated them into the pre-defined themes in the coding frame (Table S1).*"

Comment 9: That said, this paper provides keen insight into the experience of people picking up their drugs at a pharmacy instead of spending a great deal of time at a clinic to pick up medicines. It's good to see this, but much of it seems like common sense. It is hard to imagine how the results could be other than they are.

Response: Thank you for this comment.

Reviewer: 3

Comment 10: Overall this is a very well written manuscript tackling an important topic, reflections on the implementation of the chronic care dispensing program in South Africa. Although the noted barriers to clinic care were not surprising and replicate findings from other studies, the section focusing on Centralised Chronic Medication Dispensing and Distribution (CCMDD) implementation was interesting and provided new insights. Clearer highlighting in the abstract regarding findings around CCMDD implementation heterogeneity would guide readers to what I found to be the most compelling elements of the article.

Response: We thank the reviewer for their comments. We have highlighted some findings around CCMDD implementation in the abstract below:

Lines 31-32: “*Clients and healthcare workers had to negotiate problems with implementation of CCMDD, including some pharmacies reaching capacity or only allowing ART collection at restricted times.*”

Minor comments

Comment 11: Introduction, Line 69: Please clarify if anti-hypertensives are the only non-HIV drug dispensed or provide additional examples/list all (if reasonable)

Response: As suggested, we have made it clearer that the CCMDD programme provides non-HIV treatment and have provided some additional examples of medication that is available.

Lines 69-75: “*South Africa currently runs the largest differentiated ART delivery programme in the world as part of the Centralized Chronic Medication Dispensing and Distribution Programme (CCMDD). The CCMDD programme allows people with and without HIV to collect pre-packed chronic medications (e.g. ART, antihypertensives, diabetes treatment, lipid-lowering medication,) in the community, instead having treatment dispensed at clinics. People living with HIV can be referred into CCMDD by their ART clinic if they are non-pregnant, stable on ART and have had two consecutive suppressed HIV viral loads, at least six months apart.*”

Comment 12: Methods, Line 109: Based on Table 1, selection of participants including virally suppressed and non-virally suppressed appears to be based on baseline or 6-month viral loads, not viral loads at the end of the study. Please clarify in line 109.

Response: As suggested, we have clarified that viral loads from any time point in the study were used to select participants.

Lines 125-126: “*We used a purposive sampling frame to select a range of patients including those who received care in CCMDD, who had suppressed and unsuppressed viral loads at any point in the study, and who received point-of-care testing versus standard laboratory testing in the STREAM trial (Table 1).*”

Comment 13: Line 112: Please clarify if informed consent was written or verbal

Response: We clarify informed consent procedures below:

Lines 128-129: “*... and obtained written informed consent for their participation.*”

Comment 14: Data Collection: How were focus groups organized or stratified (by sex/gender, by CCMDD vs. non-CCMDD, etc) over was everyone interviewed together independent of the reasons that they were purposively sampled?

Response: Thank you for noting this. We have clarified this below:

Lines 137-138: “*Focus groups were broadly organised by STREAM study arm, and by attendance in CCMDD.*”

Comment 15: Results: Were there any differences by age or gender? The sample is well balanced at least in terms of gender, but it is not clear if there were any differences between men and women (including the gender of the individuals providing the quotes) which would be interesting to understand.

Response: We did not note any marked differences between men and women, and have provided the gender of individuals who provided quotes. We did not provide demographic details of staff to protect their confidentiality.

Comment 16: Most of the CCMD engagement info focused on those attending pharmacies. Were there any insights from those in non-pharmacy settings?

Response: Thank you for this comment. Several of the quotes included did come from non-pharmacy settings, and we have now made this clearer in the quotes below:

Line 205: *Collecting... [at the community-based organisation] saves a lot of time...*

Line 249: *“It’s quicker to just collect [at the community-based organisation], because you know how...”*

Line 263: *“Otherwise, it’s fine because it [the community-based organization] is quick. If your vital signs were done as well, it would have been better.”*

Line 403: *“Everything [at the community-based organisation] is perfect and fast. If... I’m supposed to start work at half past eight I can make it,”*

Comment 17: Because patients had to present an ID, did this mean that they could no longer send family/friends? If so, how was this perceived?

Response: To be registered in the system, patients required an ID, but were then able to nominate someone else to collect treatment for them.

Comment 18: Discussion, Line 314: assessment should be assessments

Response: Thank you for noting this, we have now corrected to ‘assessments’

VERSION 2 – REVIEW

REVIEWER	Tom Decroo Institute of Tropical Medicine Antwerp, Belgium
REVIEW RETURNED	18-Mar-2020
GENERAL COMMENTS	Dear authors, I thank you for the revised manuscript. I read it with interest. In my view coherence and clarity has improved substantially, as you restructured the manuscript taking into account the major themes. Kind regards